# Lack of Mitochondrial DNA Provides Metabolic Advantage in Yeast Osmoadaptation

**DOI:** 10.3390/biom14060704

**Published:** 2024-06-14

**Authors:** Maria Antonietta Di Noia, Ohiemi Benjamin Ocheja, Pasquale Scarcia, Isabella Pisano, Eugenia Messina, Gennaro Agrimi, Luigi Palmieri, Nicoletta Guaragnella

**Affiliations:** Department of Biosciences, Biotechnologies and Environment, University of Bari “Aldo Moro”, 70125 Bari, Italy; maria.dinoia@uniba.it (M.A.D.N.); ohiemi.ocheja@uniba.it (O.B.O.); pasquale.scarcia@uniba.it (P.S.); isabella.pisano@uniba.it (I.P.); eugenia.messina@uniba.it (E.M.); gennaro.agrimi@uniba.it (G.A.); luigi.palmieri@uniba.it (L.P.)

**Keywords:** mitochondrial DNA, yeast, stress response, osmoadaptation, metabolism, retrograde signaling, ROS

## Abstract

Alterations in mitochondrial function have been linked to a variety of cellular and organismal stress responses including apoptosis, aging, neurodegeneration and tumorigenesis. However, adaptation to mitochondrial dysfunction can occur through the activation of survival pathways, whose mechanisms are still poorly understood. The yeast *Saccharomyces cerevisiae* is an invaluable model organism for studying how mitochondrial dysfunction can affect stress response and adaptation processes. In this study, we analyzed and compared in the absence and in the presence of osmostress wild-type cells with two models of cells lacking mitochondrial DNA: ethidium bromide-treated cells (ρ^0^) and cells lacking the mitochondrial pyrimidine nucleotide transporter *RIM2* (Δ*RIM2*). Our results revealed that the lack of mitochondrial DNA provides an advantage in the kinetics of stress response. Additionally, wild-type cells exhibited higher osmosensitivity in the presence of respiratory metabolism. Mitochondrial mutants showed increased glycerol levels, required in the short-term response of yeast osmoadaptation, and prolonged oxidative stress. The involvement of the mitochondrial retrograde signaling in osmoadaptation has been previously demonstrated. The expression of *CIT2*, encoding the peroxisomal isoform of citrate synthase and whose up-regulation is prototypical of RTG pathway activation, appeared to be increased in the mutants. Interestingly, selected TCA cycle genes, *CIT1* and *ACO1*, whose expression depends on RTG signaling upon stress, showed a different regulation in ρ^0^ and Δ*RIM2* cells. These data suggest that osmoadaptation can occur through different mechanisms in the presence of mitochondrial defects and will allow us to gain insight into the relationships among metabolism, mitochondria-mediated stress response, and cell adaptation.

## 1. Introduction

Mitochondria are essential organelles whose primary role is the synthesis of ATP by oxidative phosphorylation. However, it is well known that mitochondria also control a variety of other cellular functions including amino acid, nucleotide, lipid and heme biosynthesis; the redox state; genomic stability and active stress responses ultimately determining the fate of the cell. In line with the multifaceted role of these organelles, alterations in mitochondrial function have been linked to many human pathologies ranging from rare genetic disorders to common diseases, such as neurodegeneration [1], cancer [2,3,4] and the aging process itself [5,6,7,8,9].

Mitochondrial capacity to maintain cellular energy homeostasis and cell survival relies on a robust signaling system between mitochondria and other subcellular compartments or organelles, such as the nucleus, the endoplasmic reticulum, lysosomes and peroxisomes [10,11,12].

This mito-cellular signaling involves molecular determinants ranging from metabolites to misfolded proteins or reactive oxygen species (ROS) and acts to restore cellular homeostasis and facilitate adaptation to the altered mitochondrial status or eliminate dysfunctional mitochondria via mitophagy. Cellular response to mitochondrial dysfunction, known as the mitochondrial retrograde response, has been studied in a wide number of organisms. The best-characterized mechanism of cell response to mitochondrial dysfunction is the RTG pathway in yeast *Saccharomyces cerevisiae*. RTG pathway activation leads to a reconfiguration in the expression of a subset of nuclear genes that rewires metabolism to bypass the block in the tricarboxylic acid (TCA) cycle and promote cell survival [13,14]. Metabolic adjustment is a cytoprotective strategy in stress response and it has been recently shown that mitochondria can support the survival of cancer cells upon a hyperosmotic environment [15].

Given its capacity to tolerate respiratory defects or the complete loss of mitochondrial DNA, *S. cerevisiae* is an ideal model to study how mitochondrial functionality can affect adaptation processes. In this study, we investigated whether and how the lack of mitochondrial DNA (mtDNA) can impact osmoadaptation. Yeast osmoregulation is characterized by the production of the compatible solute glycerol, which is mainly regulated by the HOG mitogen-activated protein kinase (MAPK) pathway [16,17]. We have shown that the RTG pathway is also involved in osmoadaptation acting downstream of the Hog1-mediated line of defense and sustaining mitochondrial respiratory capacity [18]. Although the involvement of mitochondria in the response to osmostress is clear, their role in terms of sensing, integrating and transducing stress signals remains to be elucidated. Recently, we have demonstrated that the inactivation of *HAP4*, encoding the catalytic subunit of the transcriptional complex Hap, accelerates osmoadaptation in an RTG-dependent fashion [19].

In this study, we analyzed and compared wild-type cells with ethidium bromide-treated cells (ρ^0^) and cells lacking *RIM2* in the presence or absence of NaCl. It is well known that ethidium bromide treatment interferes with mitochondrial DNA replication, generating ρ^0^ cells. On the other hand, Rim2p is a member of the mitochondrial carrier protein family and has been identified as a yeast mitochondrial pyrimidine nucleotide transporter. The main physiological role of Rim2p is to transport (deoxy)pyrimidine nucleoside triphosphates into mitochondria in exchange for intra-mitochondrially generated (deoxy)pyrimidine nucleoside monophosphates [20]. The deletion of *RIM2* causes total loss of mtDNA and lack of growth on non-fermentative carbon sources. Furthermore, Δ*RIM2* exhibited a fragmented mitochondrial morphology similar to that of wild-type ρ0 cells [21,22].

Overall, our data show that strains lacking mitochondrial DNA have a common advantage in the kinetics of stress response compared to wild-type cells, but different mechanisms of stress signaling might be involved. Evidence is reported for a role of mitochondria as metabolic hubs rather than bioenergetic ones in cellular adaptation to osmostress.

## 2. Materials and Methods

### 2.1. Yeast Strains and Growth Conditions

The *S.cerevisiae* strains used in this study were *W303-1B* (WT) cells (MATα ade2 leu2 his3 trp1 ura3) and derivatives ρ^0^, ∆*HAP4* (∆*HAP4*::KanMX4) and Δ*RIM2* obtained as described in [18,22], respectively.

Cells were grown at 30 °C in a rich YP medium containing 1% yeast extract, 2% bactopeptone (GIBCO, Life Technologies, Waltham, MA, USA) and 2% glucose (YPD) or 2% galactose (YPG) or 2% ethanol (YPE), in the absence or in the presence of 0.8 M sodium chloride (NaCl). Cell growth was monitored quantitatively by measuring optical density (600 nm) on liquid cultures grown either in micro-well plates or in flasks. Glucose, galactose, ethanol and NaCl have been provided from Sigma-Aldrich, St. Louis, MO, USA.

### 2.2. Micro- and Batch-Culture Growth Assays

For micro- and batch-culture growth assays, fresh yeast cells cultured for about 16 h at 30 °C were diluted in triplicate in multi-well plates or flasks to the same initial OD600. Optical density was then constantly monitored using a high-precision TECAN Sunrise microplate reader equipped with a shaker and a temperature control unit or by using a Thermo Spectronic Genesis20 spectrophotometer (Analytical control, s.p.a. Cinisiello Balsamo, Milano, Italy) at selected times. Micro-culture growth curves were analyzed in Microsoft Excel and doubling time was calculated as reported in Toussaint et al. [23]. Relative growth was calculated in both micro- and batch-cultures as the percentage of the optical density values under stress conditions (with NaCl) compared to the control (without NaCl) at different times. At least three independent cultures were analyzed for each condition in each independent experiment.

### 2.3. High-Performance Liquid Chromatography (HPLC) Analysis

Fresh yeast cells cultured for about 16 h at 30 °C were diluted to 0.1 OD600 in fresh liquid YPD with or without 0.8 M sodium chloride. After 12 and 24 h, cells were centrifuged and the supernatants were diluted 1:1 with mobile phase H_2_SO_4_. Extracellular glucose, ethanol and glycerol were identified and quantified by HPLC, using a Waters Alliance 2695 separation module (Waters, Milford, MA, USA) equipped with a Resex ROA-Organic Acid H+ (8%) 300 mm × 7.8 mm column (Phenomenex Inc., Torrance, CA, USA), coupled to a Waters 2410 refractive index detector and a Waters 2996 UV detector. Separation was carried out at 60 °C with 0.0025 M H_2_SO_4_ as a mobile phase at a flow rate of 0.5 mL/min. Samples were identified by comparing the retention times with those of standards.

### 2.4. Flow Cytofluorimetry Analysis

Cells were collected at appropriate intervals and diluted to a final optical density of 0.1. ROS production and membrane integrity were analyzed with Attune™ NxT Acoustic Focusing Cytometer (Thermo Fisher, Waltham, MA, USA) by using specific fluorescent dyes. 2′,7′-Dichlorofluorescein diacetate, DCF-DA (Sigma-Aldrich, St. Louis, MO, USA) was used as a probe for the detection of intracellular ROS levels. Yeast cells were stained with 50μM DCF-DA and incubated at 30 °C in the dark for 30 min. After washing with PBS, the green fluorescence of cells was measured in terms of the percentage of DCF-positive cells to assess the ROS levels. Membrane integrity was investigated with propidium iodide (PI) (Sigma-Aldrich Inc.), a red fluorescent DNA intercalator, that penetrates membrane-damaged yeast cells. Cells were incubated with 46 μM PI at room temperature in the dark for 15 min. A flow cytometric analysis used an exciting wavelength of 488 nm and emission wavelength of 530/30 nm for green fluorescence detection (DCF-DA), 695 ± 40 nm (BL-3) and 574 ± 26 nm (BL2), for red and orange fluorescence, respectively (PI).

### 2.5. Quantitative PCR (qPCR)

The mRNA levels of a peroxisomal citrate synthase-encoding gene (*CIT2*), a mitochondrial citrate synthase-encoding gene (*CIT1*) and aconitase (*ACO1*) were determined in continuously growing cells after 5 h of NaCl exposure and in the absence of stress. A total of 5 × 10^7^ cells were collected and centrifuged at 3000 g. Cell pellets were stored at −80 °C before total RNA extraction with a Presto Mini RNA Yeast Kit (Geneaid, New Taipei City, Taiwan). We immediately performed the cDNA synthesis preceded by DNase treatment, using Superscript IV VILO Mastre Mix (Invitrogen, Thermo Fisher Scientific, Waltham, MA, USA) according to their standard protocols. cDNA was directly used for the quantitative PCR (qPCR) analysis or stored at −20 °C. The QuantStudio 3 Real-Time PCR System (Applied Biosystems, Thermo Fisher Scientific, Waltham, MA, USA) was used for Quantitative PCR using the primer pairs (Table 1) based on the cDNA sequences of the investigated genes and designed with Primer Express 3.0 (Applied Biosystems, Thermo Fisher Scientific, Waltham, MA, USA). The primers were purchased from Invitrogen (Thermo Fisher Scientific, Waltham, MA, USA). Twenty microliters of reaction volume contained 20 ng of reverse-transcribed first-strand cDNA, 10 μL of SYBR Select Master Mix (Applied Biosystems, Thermo Fisher Scientific, Waltham, MA, USA), and 300 nM of each primer. The specificity of the PCR amplification was checked with the heat dissociation protocol after the final cycle of PCR.

To correct for differences in the amount of starting first-strand cDNAs, the *ACT1* mRNA was amplified in parallel as a reference gene. The relative quantification of the investigated genes was performed according to the comparative method (2^−ΔΔCt^) [24]; 2^−ΔΔCt^ = 2 − ^(ΔCtsample − ΔCtcalibrator),^ where ΔCt sample is the Ct sample − the Ct reference gene and Ct is the threshold cycle. For the calibrator, ΔΔCt = 0 and 2^−ΔΔ^ Ct =1. The value of 2^−ΔΔ^ Ct indicates the fold change in gene expression relative to the calibrator (wild-type cells grown without NaCl stress).

### 2.6. Statistical Analysis

All the experiments were repeated at least three times, and the results are reported as means with standard deviation. Student’s t-test was used to determine significant differences between samples, using Microsoft Excel software (Microsoft® Excel® per Microsoft 365 MSO, Versione 2405 Build 16.0.17628.20006) with significant differences reported as *p*-values between 0.05 and 0.0001 for all results, except for qPCR data, where one-way ANOVA followed by Tukey’s multiple comparison test was used.

## 3. Results

### 3.1. Cells Lacking Mitochondrial DNA Exhibit Faster Osmoadaptation

Mitochondria are key components of the stress response, which might differ depending on genetic and/or environmental factors that determine mitochondrial function. Therefore, to study whether the lack of mtDNA could impact the osmostress response, ρ^0^ and Δ*RIM2* cells were compared to WT and Δ*HAP4*, lacking the catalytic subunit of the transcriptional complex Hap in the presence and in the absence of NaCl in a rich medium with glucose as the sole carbon and energy source [19]. The concentration of 0.8 M NaCl causing mild salt stress was used as in [18]. Cell growth was analyzed in both micro- and macro-cultures (shake flask) up to 24 h and relative growth calculated for each sample (Figure 1A and Figure 1B, respectively). The relative growth of ρ^0^ and Δ*RIM2* was higher compared to WT at each time analyzed.

After 24 h, the relative growth of ρ^0^ and Δ*RIM2* was about 80% versus 60% of WT (Figure 1A). As expected, data from micro-cultures confirmed a higher relative growth of Δ*HAP4* cells compared to WT, but lower compared to ρ^0^ and Δ*RIM2* ([19] and Figure 1A). The increase in doubling time upon NaCl exposure was significantly smaller for ρ^0^ (3.2 h) and Δ*RIM2* (3.6 h) compared to WT (4.3) (Table 2). Results from shake flask cultures were comparable to micro-cultures, confirming a higher relative growth of ρ^0^ and Δ*RIM2* compared to WT upon stress. It is worth noting that in these conditions, the improved growth phenotype of Δ*RIM2* was observed earlier than in ρ^0^ and Δ*HAP4*, with a relative growth of about 70% at 16 h. At 24 h, the mutants showed a significant increase compared to WT (Figure 1B). In the same conditions, ρ^0^ cells require a longer stress recovery time with slower kinetics of osmoadaptation compared to Δ*RIM2*, but adaptation was improved after 24 h, when the relative growth reached 62% compared to 48% of WT cells.

These results demonstrate that the lack of mtDNA accelerated osmoadaptation.

### 3.2. The Kinetics of Osmoadaptation Is Affected by Metabolic Conditions

Since it is well known that the absence of mitochondrial DNA abolishes respiratory capacity, we evaluated whether the modulation of respiratory metabolism could affect the kinetics of osmoadaptation. Thus, WT cells were grown in galactose or ethanol as sole carbon sources, activating a respiro-fermentative and a fully respiratory metabolism, respectively, and compared for growth. Relative growth was first assessed in shaking flasks in the presence of 0.8 M NaCl at 24 h, showing a value of about 20% in galactose or ethanol compared to 50% measured in glucose (Figure 2A). Since no significant differences could be observed between respiro-fermentative and fully respiratory conditions, milder NaCl stress (0.4 M) was tested along time. In these conditions, NaCl did not affect relative growth of WT cells grown in glucose, while a value of about 80% and 60% was observed for galactose and ethanol, respectively, after 24 h.

These results indicate that an active mitochondrial metabolism negatively affects the response to osmotic stress. This is in accordance with the evidence that cells lacking mitochondrial DNA, with the absence of respiration and a known high ratio of NADH/NAD+, would have a selective advantage in the kinetics of osmoadaptation.

### 3.3. The Production of Glycerol upon Osmostress Is Increased in Cells Lacking Mitochondrial DNA

It is well established that the first line of cell response to osmotic stress is the synthesis of glycerol activated by the High Osmolarity Glycerol (HOG) pathway [17] and references therein. Glycerol is produced in a two-step pathway from the glycolytic intermediate dihydroxyacetone phosphate (DHAP); therefore, this signaling pathway controls the redistribution of the glycolytic flux from growth to glycerol production. Since cells lacking mtDNA rely on a fermentative metabolism, we hypothesized that glycolytic flux was directed towards the production of the osmolyte glycerol favoring faster adaptation to salt stress. Thus, ethanol and glycerol productions were measured by HPLC and compared among WT and mutant cells treated or not with NaCl (Figure 3). In the absence of stress, the cells lacking mtDNA showed a significant accumulation of extracellular ethanol, as a product of fermentative metabolism, compared to WT and Δ*HAP4* at 24 h (Figure 3A). On the other hand, in the presence of NaCl, no significative differences of extracellular ethanol were observed among the strains. It is notable that cell growth is delayed in the presence of NaCl and consequently the rate of glucose consumption decreased in all strains, affecting ethanol production. Differently to ethanol, glycerol production was about three-fold higher in ρ^0^ and Δ*RIM2* compared to WT and Δ*HAP4* in the absence of stress at both 12 and 24 h, while NaCl treatment induced a glycerol increase, although to a different extent, in all cell types with no significant differences (Figure 3B).

These data demonstrate that cells lacking mitochondrial DNA produced a higher basal level of glycerol. This, in accordance with a more active fermentative metabolism and consequent increase in NADH levels [25], could improve the kinetics of osmoadaptation.

### 3.4. ROS Production Is a Common Trait of Osmostress Signaling

It is well known that reactive oxygen species (ROS) may act as an adaptive signal in response to various stressors; thus, the levels of ROS were analyzed in the absence and in the presence of NaCl treatment in WT and mutant cells along time. The fluorescent dye DCFH-DA sensitive to intracellular oxidation was used for the assay. In WT cells, the percentage of DCF-positive cells was about 60% and 20% at 6 and 12 h, respectively, in the absence of stress; NaCl caused a slight increase at each time, but it is of note that a decrease in ROS accumulation was observed at 12 h in control and treated cells. A high percentage of positive cells was observed in mitochondrial mutants already at 6 h, independently of NaCl stress. At 12 h, a decrease was observed in all mutants in the control conditions. In the presence of NaCl, at 12 h, ρ^0^ and Δ*RIM2* maintained an elevated ROS content, while Δ*HAP4* showed a decreased ROS production similarly to WT (Figure 4).

These data confirm that NaCl treatment increased ROS production and demonstrate that prolonged oxidative stress occurred in the absence of mtDNA.

### 3.5. RTG Pathway and TCA Cycle Genes Are Differently Regulated in ρ^0^ and ΔRIM2 Cells

It has been previously shown that the RTG pathway plays an important role in osmoadaptation by sustaining mitochondrial function [18,19]. In order to gain insight into the mechanisms determining the faster osmoadaptation in cells lacking mtDNA, the expression of *CIT2*, the prototypical target gene of RTG pathway activation, was assessed in ρ^0^ and Δ*RIM2* cells in the absence and presence of NaCl. As expected, *CIT2* basal mRNA levels were higher in ρ^0^ unstressed cells compared to WT, while only a slight increase was observed for Δ*RIM2* (Figure 5). Osmostress elicited *CIT2* up-regulation in all cell types, although with different extents: about a 5-fold and a 2-fold increase for WT and ρ^0^, respectively, compared to no-stress conditions. Interestingly, in Δ*RIM2* cells, only a slight increase in *CIT2* mRNA could be observed in the presence of NaCl. The RTG pathway was activated similarly to WT in Δ*HAP4* cells [19].

These results further confirm the relevance of the RTG pathway in the response to osmostress, but suggest its differential activation and possible regulation in yeast cell models lacking mtDNA.

When yeast cells lack functional mitochondria, the expression of TCA cycle enzymes leading to the synthesis of α-ketoglutarate is controlled by the RTG pathway [26,27]. We have previously shown that in faster osmoadapted cells lacking *HAP4*, the first genes of the TCA cycle were up-regulated with a specific RTG-dependent activation of *CIT1*, encoding mitochondrial citrate synthase 1 [19]. Thus, the levels of expression of *CIT1* and *ACO1*, encoding aconitase 1, were measured in ρ^0^ and Δ*RIM2* cells and compared to WT, in the presence and in the absence of NaCl (Figure 6). It was evident that the expression of these genes was differently regulated in all cell types. Particularly, in the absence of stress (Figure 6, white bars), no significant variations could be observed among the strains, except for *CIT1*, whose expression appeared to be much lower in Δ*RIM2* cells compared to both WT and ρ^0^. NaCl treatment (Figure 6, black bars) resulted in the up-regulation of *ACO1*, but not of *CIT1*, in ρ^0^ cells and in a slight down-regulation of *CIT1* in Δ*RIM2* cells. Under both basal and stress conditions, the expression of *CIT1* and ACO1 measured in ρ^0^ cells was always higher than the expression measured in Δ*RIM2* cells. The down-regulation of *CIT1* and *ACO1* was confirmed in WT cells upon osmostress as in [19].

These data reveal that the first TCA cycle genes could play a role in the kinetics of osmoadaptation and define differences in the metabolic state of ρ^0^ and Δ*RIM2* cells.

## 4. Discussion

Mitochondrial dysfunction can be considered from a dual perspective in cellular stress response. In fact, stress insults can alter mitochondrial functionality and lead to cell death, but meanwhile mitochondrial dysfunction itself can activate pro-survival pathways favoring cell adaptation. In this work, we chose yeast *Saccharomyces cerevisiae* as a model organism to investigate the impact of mitochondrial DNA absence on adaptation to osmotic stress. In particular, ethidium bromide-treated cells (ρ^0^) and cells lacking the mitochondrial pyrimidine nucleotide transporter *RIM2* have been selected for the analysis and reciprocal comparison with respect to wild-type and Δ*HAP4* cells.

Cellular growth, ethanol and glycerol production, oxidative stress, activation of the RTG pathway and expression of key TCA cycle genes have been assessed along time in the presence and in the absence of NaCl stress. Our results demonstrate that the lack of mtDNA confers an advantage in the kinetics of osmoadaptation, which appeared faster in the mutants with respect to wild-type cells (Figure 1 and Table 1). What is the molecular basis for this advantage? It is well known that the first line of cellular response to osmotic stress is the biosynthesis of glycerol deriving from glucose, whose flux is mainly redirected from biomass towards glycerol production [28]. This is confirmed from our data, where all the cells treated with NaCl showed an increased production of glycerol compared to control conditions. In addition, the levels of glycerol in ρ^0^ and Δ*RIM2* cells were significantly higher than in the wild type and Δ*HAP4* in the absence of stress (Figure 3B), highlighting the ability of cells lacking mitochondrial DNA to secrete more glycerol in the medium [25]. The differences in glycerol levels observed between Δ*HAP4* and the other two mitochondrial mutants might be due to the ratio of NADH/NAD+. In ρ^0^ and Δ*RIM2* cells, with complete lack of respiratory capacity, the NADH/NAD+ ratio is increased and glycerol serves as a sink for the excess NADH produced and for NAD+ regeneration that supports proliferation. Instead, cells lacking *HAP4* are able to respire, at least until a certain specific growth rate, and in synergistic action with the RTG pathway (data not shown and [29]). Therefore, impaired NADH oxidation is less likely to trigger a massive glycerol production in Δ*HAP4*. This is also in agreement with the higher extracellular accumulation of ethanol in ρ^0^ and Δ*RIM2* cells compared to Δ*HAP4* in the absence of stress (Figure 3A). On the other hand, the quantification of ethanol yield indicates a sustained glycolytic flux upon osmostress in all the cells treated with NaCl with no significant differences (Figure 3A), suggesting that the first line of stress response known to be mediated by *HOG1* is a common mechanism [30].

It has been previously shown that environmental growth conditions can result in different responses to osmotic stress [31]. Our data further clarify this point, demonstrating that the metabolic state, such as a fully respiratory or a respiro-fermentative metabolism, can increase the sensitivity of the cells to osmostress, with the lowest growth observed in the presence of a mild NaCl treatment, that is, 0.4 M, and ethanol as a sole carbon source (Figure 2). This observation, together with the faster kinetics of osmoadaptation found in the presence of mitochondrial dysfunction, led us to propose that a respiratory metabolic state might be somehow detrimental for cell stress adaptation. Similarly, an integrated stress response (ISR) activated by mitochondrial dysfunction has been reported in mammalian cells relying on a glycolytic metabolism [32,33,34]. Moreover, considering the mitochondrial mutants analyzed in this work, it is possible to assert that the advantage in osmoadaptation does not depend on mitochondrial respiratory capacity. Indeed, the absence of respiration in cells lacking mtDNA favors the direction of the metabolic flux toward the fermentation pathway.

The analysis of the ROS level in NaCl-treated cells confirmed that oxidative stress is a trait of osmostress response (Figure 4), in accordance with [35]. However, a different modulation has been found along time for the mitochondrial mutants, supporting a complex correlation between respiratory capacity/metabolic conditions, ROS and stress response [36,37]. The involvement of the RTG pathway, which could orchestrate a hormetic antioxidant response upon stress conditions, might also be taken into consideration [38,39].

In the adaptation process activated by the mitochondrial mutants, the RTG pathway plays a major role, acting in a second line of stress defense for the reconfiguration of metabolism [18]. Here, we demonstrated that the RTG pathway is induced by osmotic stress also in ρ^0^ cells, where already a basal level of *CIT2* expression is higher than in the wild type (Figure 5) as already reported in [13]. Interestingly, in Δ*RIM2* cells, only a slight up-regulation of *CIT2* has been observed, indicating the involvement of different retrograde signaling. Also, the TCA cycle genes, *CIT1* and *ACO1*, which have been previously shown as putative key checkpoints in the metabolic rewiring upon osmostress, appeared to be differently expressed in the mutants. Particularly, *CIT1* levels were almost unchanged by NaCl treatment in ρ^0^ cells, while they were significantly down-regulated in Δ*RIM2* cells, even in the absence of stress (Figure 6). We had previously reported that *CIT1* was up-regulated upon osmostress in Δ*HAP4* cells and the sustained expression of *CIT1* observed in ρ^0^ cells can be explained by the repression of *HAP4* in these cells [19,40]. These results reinforce the relevance of peroxisomes–mitochondria crosstalk through the RTG pathway and possibly of citrate levels as signaling molecules in osmoadaptation [18].

The differences in gene expression observed for Δ*RIM2* cells suggest that an RTG-independent signaling might be activated in the presence of osmostress and mitochondrial dysfunction [41,42]. In this respect, it is worth mentioning that Rim2p has been reported to be required for iron utilization in mitochondria, as, for example, in Fe-S protein maturation and heme synthesis [43] and can be able to import iron and other divalent metal ions into the mitochondria in co-transport with pyrimidine nucleotides [44]. Hence, it is possible to hypothesize that the iron sulfur cluster (ISC) biosynthesis or heme formation could be altered in Δ*RIM2* cells with an iron signaling cascade activated.

This work provides new experimental evidence on the mechanisms of stress adaptation occurring in the presence of mitochondrial dysfunction. It is not a question of mitochondrial dysfunction itself, but how different mitochondrial dysfunction can activate distinct cell signaling remodeling for survival. There is not a unique mechanism governing the response, but rather many pathways whose activation depends on the nature of the mitochondrial defect and on the metabolic state of the cell. Attention should be paid to metabolic clues, which can potentially guide distinct mechanisms of adaptation. Understanding the interplay between stress mediators and environmental context will help to identify key points of reconfiguration in the dynamics of cell stress adaptation. This will represent a resource of information also for potential therapeutic targets for many stress- and mitochondria-related pathologies.

## Figures and Tables

**Figure 1 biomolecules-14-00704-f001:**
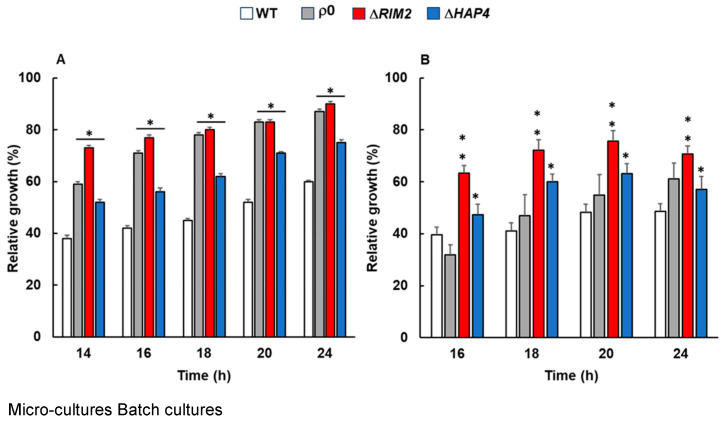
Relative growth of wild-type and mutant cells in the presence of osmostress. Wild-type (WT) and indicated mutant cells (ρ^0^, Δ*RIM2*, Δ*HAP4*) grown overnight in YPD were diluted to (**A**) 0.01 OD600 in fresh liquid YPD with or without 0.8 M NaCl and optical density was measured at 600 nm (OD600) over time with a high-precision TECAN microplate reader or (**B**) were diluted to 0.1 OD600 in fresh liquid YPD batch cultures with or without 0.8 M NaCl and optical density (OD600) was measured at the indicated times. In both (**A**,**B**), relative growth was calculated as the percentage of the OD600 of stressed/control cells. Unpaired Student’s *t*-test: a statistically significant difference with * *p* < 0.01; ** *p* < 0.001 when comparing wild-type versus mutant cells from three independent experiments.

**Figure 2 biomolecules-14-00704-f002:**
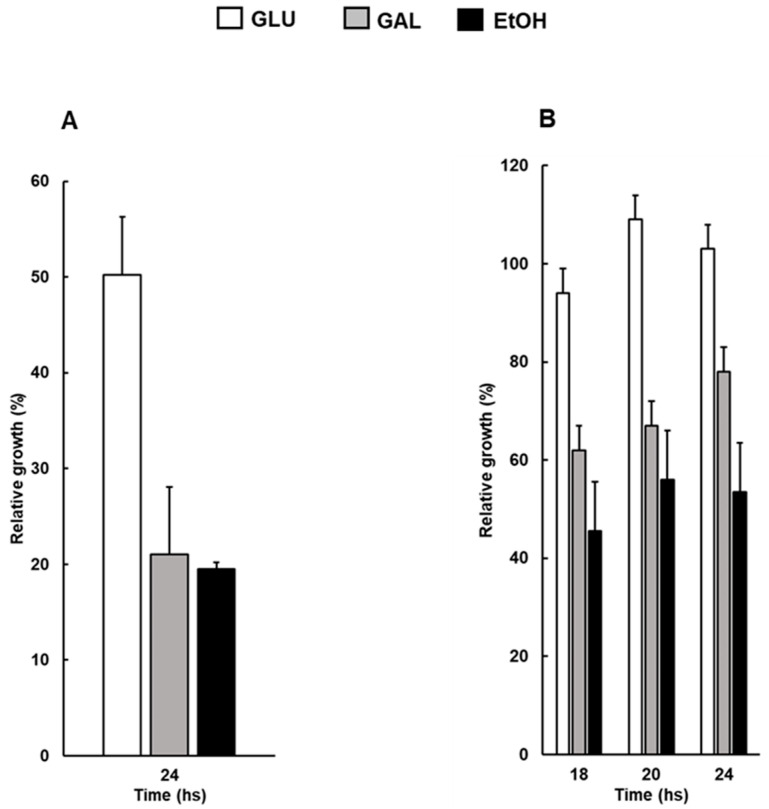
Osmoadaptation of wild-type cells grown in different carbon sources. (**A**) Wild-type (WT) cells, grown overnight in YPD, were diluted in fresh liquid YP containing glucose (white bars), galactose (grey bars) or ethanol (black bars), with or without sodium chloride (NaCl) at 0.8 M (**A**) or 0.4 M (**B**) and optical density was measured at 600 nm (OD600) at indicated hours. In all experiments, relative growth was calculated as percentage of OD_600_ of stressed/control cells.

**Figure 3 biomolecules-14-00704-f003:**
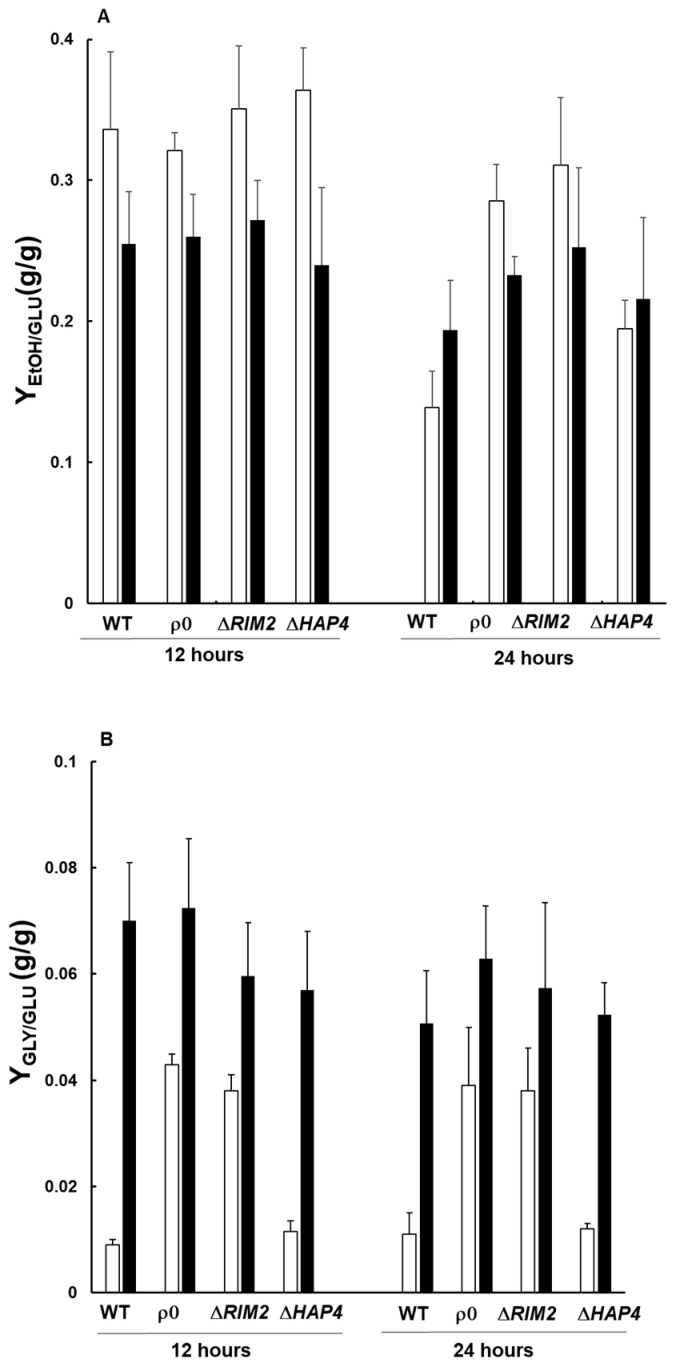
Yield of ethanol and glycerol in the absence and presence of osmostress. Wild-type (WT) and mutant cells grown overnight in the YPD medium were diluted to 0.1 OD600 in fresh liquid YPD with (black bars) or without (white bars) 0.8 M sodium chloride (NaCl). After 12 and 24 h, cells were centrifuged and the supernatants were diluted 1:1 with mobile phase 2.5 mM H_2_SO_4_ to measure extracellular ethanol (**A**) and glycerol (**B**) by HPLC.

**Figure 4 biomolecules-14-00704-f004:**
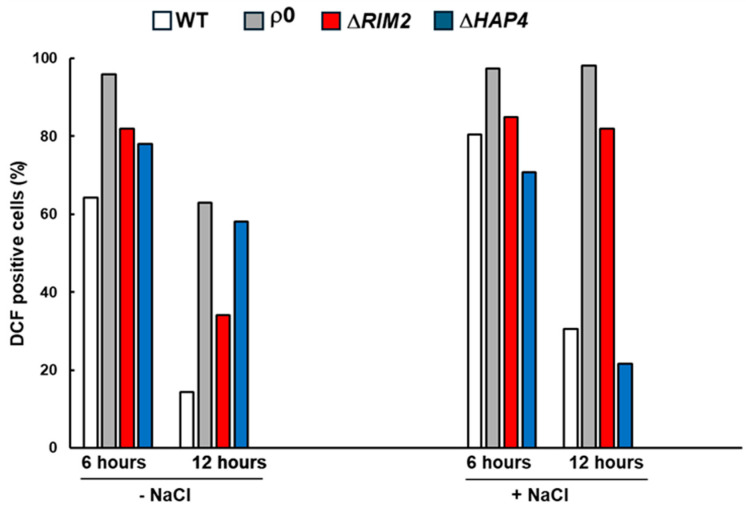
Effect of osmotic stress on ROS production. Yeast cells WT, ρ^0^, Δ*RIM2*, and Δ*HAP4* were grown in YPD medium in absence and in presence of NaCl at 0.8 M and intracellular ROS level was measured at indicated times. Data reported refers to representative experiment from two independent biological replicates.

**Figure 5 biomolecules-14-00704-f005:**
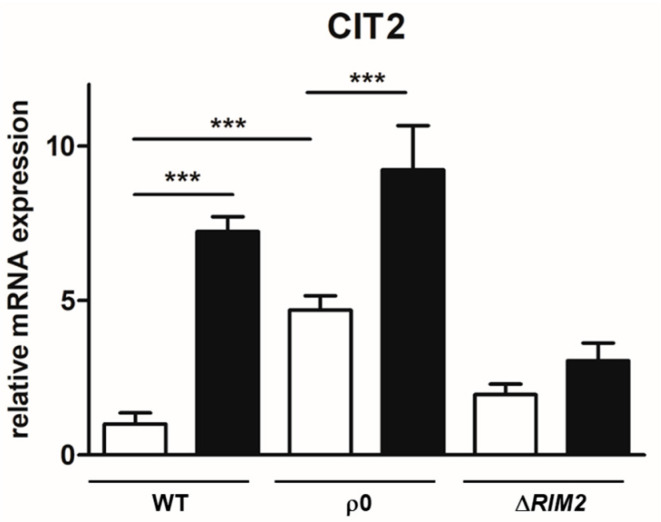
*CIT2* expression under high-osmotic environment. Wild-type (WT) and mutant cells (ρ^0^, Δ*RIM2*) grown overnight in YPD medium were diluted to 0.1 OD600 in fresh liquid YPD with (black bars) or without 0.8 M sodium chloride (white bars). After 5 h, cells were collected for RNA extraction, and mRNA levels were measured by quantitative PCR. Relative amount of *CIT2* mRNA was calculated according to comparative method (2^−ΔΔCt^) and wild-type cells grown without NaCl stress were used as calibrator. Values are mean ± SD of three independent experiments. One-way ANOVA, Tukey’s (post hoc) test; *** *p* < 0.001.

**Figure 6 biomolecules-14-00704-f006:**
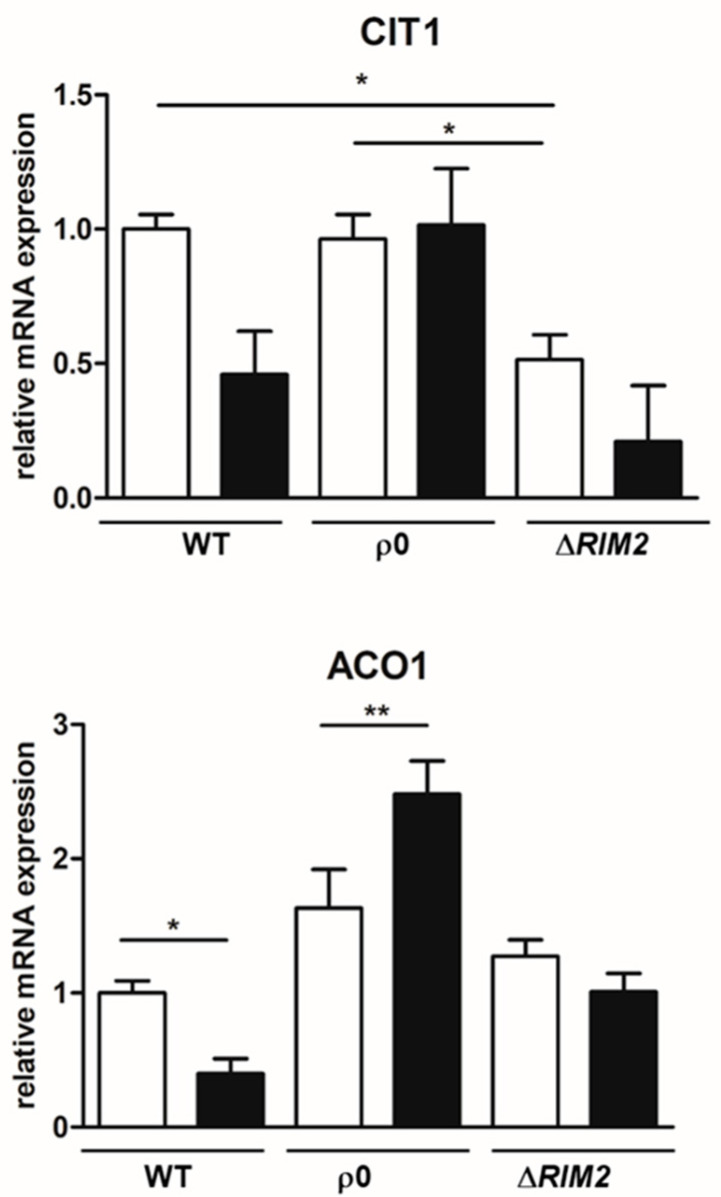
*CIT1* and *ACO1* expression under high-osmotic environment. Wild-type (WT) and mutant cells (ρ^0^ and Δ*RIM2*) grown overnight in YPD medium were diluted to 0.1 OD600 in fresh liquid YPD with (black bars) or without 0.8 M sodium chloride (white bars). After 5 h, cells were collected for RNA extraction, and mRNA levels were measured by quantitative PCR. Relative amount of *CIT1* and *ACO1* mRNA was calculated according to comparative method (2^−ΔΔCt^) and wild-type cells grown without NaCl stress were used as calibrator. Values are mean ± SD of three independent experiments. One-way ANOVA, Tukey’s (post hoc) test; * *p* < 0.05, and ** *p* < 0.01.

**Table 1 biomolecules-14-00704-t001:** Primers used for quantitative PCR.

Primer	Sequence
CIT1 Forward	5-GCGCCTCCGAACAAACG-3
CIT1 Reverse	5-CTGCCTTTGCTGGGATAATTTC-3
ACO1 Forward	5-CAAGAACCCAGCTGACTATGACA-3
ACO1 Reverse	5-CCAATTCAGCTAGACCCAGAATATC-3
CIT2 Forward	5-TGTAAGGCAATTCGTTAAAGAGCAT-3
CIT2 Reverse	5-CCCATACGCTCCCTGGAATAC-3
ACT1 Forward	5-ACTTTCAACGTTCCAGCCTTCT-3
ACT1 Reverse	5-ACACCATCACCGGAATCCAA-3

**Table 2 biomolecules-14-00704-t002:** Doubling time (DT) of wild-type (WT) and mutant cells grown in YPD in the absence (control) and in the presence of 0.8 M NaCl. Analyses were performed in TECAN microplate reader. Reported data are the mean ± standard deviation of three independent experiments, each performed in triplicate.

Strains	DT(h) CTRL	DT(h) + NaCl
WT	2.0 ± 0.42	4.3 ± 0.57
ρ^0^	2.4 ± 0.12	3.2 ± 0.20
Δ*RIM2*	2.3 ± 0.35	3.6 ± 0.35
Δ*HAP4*	2.4 ± 0.16	3.5 ± 0.71

## Data Availability

Raw data from this study are available upon reasonable request from the corresponding author.

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
