# Peer review of "Lack of Mitochondrial DNA Provides Metabolic Advantage in Yeast Osmoadaptation"

_biomolecules, 2024, doi:10.3390/biom14060704_

Round 1
Reviewer 1 Report
Comments and Suggestions for Authors
This study follows up on previous work from the authors suggesting that alterations in mitochondrial function can facilitate osmoadaptation. Here, they focus on loss of respiration, and the data support a model in which increased production of glycerol facilitates osmoadaptation. This makes sense to me, since the lack of respiration should leave more glycerol available to act as an osmoprotectant. There is also some new data on ROS generation, which seems somewhat less relevant.
My only major concern is that the rho0 and rim2∆ mutants used here lack mtDNA, which causes defects beyond simply a loss of respiratory activity. This is manifested as smaller colony size of rho0 mutants compared to “pet” mutants in which a single nuclear gene required for respiration (but not mtDNA stability) is deleted. The authors do use changes in carbon source to alter metabolism in WT cells, but this will change many aspects of metabolism. I think it would be helpful to analyze a mutant like cox9∆ that maintains mtDNA but cannot respire.
Other concerns:
Line 249-250: This may reflect my ignorance, but I would have thought that cells lacking mtDNA exclusively rely on a fermentative metabolism, not “predominantly”. Are the authors suggesting that some respiration is possible in the absence of mtDNA?
It is my understanding that as part of the response to hyperosmotic conditions, cells increase in size. This should affect the apparent optical density of the cultures, which was used to determine growth rates. Thus, it would be appropriate to use a direct method of determining proliferation rate, such as counting cells/mL with a hemacytometer, for at least one experiment, to verify that the observed changes do in fact reflect changes in proliferation and not a composite of changes in proliferation and changes in size.
How do the authors know that the increase in CIT2 expression in the presence of NaCl is a result of RTG? An important control would be to treat a mutant defective in RTG with NaCl and assess CIT2 expression. The same caveat applies to the changes in CIT1 and ACO1 expression shown in Figure 6.
What is the source of ROS in rho0 mutants? Is this known? It would seem relevant to include this information here.
Minor suggestions:
Abstract: “In this study, we analyzed and compared wild type cells with ethidium bromide-treated cells (rho0)”: The average reader may not know that ethidium bromide interferes with mtDNA replication and generates rho0 cells, or even what “rho0” refers to (cells lacking any mtDNA). It would be best to explain/clarify this here in the Abstract, and perhaps in more detail in the Introduction.
Also, the sentence: “Accordingly, wild type cells exhibited higher osmosensitivity in the presence of respiratory metabolism”. Here, “accordingly” seems inappropriate, since it means “consequently” or at least implies that a second thing would be logically expected on the basis of the first. A defect in respiratory metabolism is not the only cellular defect in rho0 cells, so it is by no means a given that wild-type cells undergoing respiratory metabolism would be more osmosensitive than fermenting wild-type cells. I recommend replacing “Accordingly” with “Furthermore” or “Additionally”.
Line 57: “since its the capacity” not sure what is intended here.
Line 183: For ease of reading, I recommend a line break between “ analyzed.” and “After 24 hours,”
Line 184: “As expected, data from microcultures”: this sentence seems to imply that the data in question is new, and agrees with expectations based on previous data (from macrocultures). However, at the end of the sentence is a citation of the previous study of HAP4 in osmosensitivity (“[19]”), which makes it sound as if the data from microcultures in question was published in that earlier paper. Instead of the citation, should there be a callout to Figure 1B?
Figure 1: it would help the reader if rather than having to look in the figure legend to determine the difference between panels A and B, the authors could add labels like “macrocultures” and “microcultures” above the plots.
Line 187: the use of “decrease” here is a bit confusing, since the earlier part of the sentence says “in the absence and presence of stress”. The authors presumably mean that the doubling time is shorter for the mutants compared to WT, not that the doubling time is shorter in the presence of stress than in its absence. Also, “upon NaCl exposure” is more precise but is a bit redundant with the earlier “in the absence and presence of stress”. I recommend rewording to “The increase in doubling time upon NaCl exposure was significantly smaller for ρ0 (3.2 h) and ∆RIM2 (3.6 h) compared to WT (4.3) (Table 2)”.
Comments on the Quality of English LanguageThere are some minor corrections to the English here to be made. I tried to make a few suggestions to increase clarity.
Reviewer 2 Report
Comments and Suggestions for Authors
In this paper, the authors have addressed how mitochondria and/or the TCA cycle influence stress resistance. They have compared the wild type to a rho0 strain, a rim2del and a hap4del. They have analyzed the effect on growth rate on 3 media: fermentative, respiro-fermentative and respiratory. They also tested the effect of a saline test in these conditions. They also analyzed the expression levels of several key genes encoding enzymes of the carbon metabolism.
The research design is very adapted to address this question but suffers from a fatal flaw, the wild type they are using is not wild type since wild type W303 grows with a generation type of 1h30 as for example measured in https://journals.plos.org/plosone/article?id=10.1371/journal.pone.0119807. Hence no proper conclusions can be made.
Some stylistic remarks. The authors should check that the 0 is always superscripted in rho0. Are the main text and the figure legends in the same font because the rho symbol in line 193 is very different from the one in line 203. In yeast, mutant are always in lowercase and italicized and gene names capitalized and italicized. Genotypes are also italicized.
In legends of figure 1 and 2, it is not clear what the authors show since they indicate that they compare with and without NaCl yet only show one condition in panel A and in panel B. Is panel A meant to be without and B with?
Round 2
Reviewer 1 Report
Comments and Suggestions for Authors
I am satisfied by tye authors' responses to my concerns and the associated revisions to the manuscript.
Reviewer 2 Report
Comments and Suggestions for Authors
The authors' response to my first comment is iin general true, growth rate of strains will vary depending on many different factors but in the case of mitochondrial mutants it cannot be the case. Indeed rho- or rho0 mutants are precisely called petite because they form smaller colonies on YPD hence hteir growth rate is affected compared to a wild type in a significant way. Thus a wild type strain (or strain of reference) by definition has to grow faster than the mutant. If this is not the case, that means that this reference strain has a mutation that slows its growth and may and may not be related to mitochondrial function but in any case prevents a proper interpretation of the results however well the experiments are carried out.